# The Competitive Interaction of Alveolar Wall Distention with Elastin Crosslinking: A Mechanistic Approach to Emergent Phenomena in Pulmonary Emphysema

**DOI:** 10.3390/cells14100702

**Published:** 2025-05-12

**Authors:** Jerome Cantor

**Affiliations:** School of Pharmacy and Allied Health Sciences, St John’s University, Queens, NY 11439, USA; jocantor1@gmail.com

**Keywords:** pulmonary emphysema, elastin, desmosine, hyaluronan, emergent phenomena

## Abstract

Emergent phenomena arise from the interaction of competing forces at multiple scale levels, resulting in complex outcomes that are not readily apparent from analyzing the individual components. Regarding biological systems, when a critical threshold is reached, a phase transition occurs, producing a spontaneous system reorganization characterized by recognizable molecular, microscopic, and macroscopic changes. The current paper explores the emergent phenomena underlying the pathogenesis of pulmonary emphysema, a disease characterized by progressive airspace enlargement. The competitive relationship between mechanical strain imposed on alveolar walls and a countervailing increase in elastin crosslinking to prevent alveolar wall rupture leads to airspace enlargement as the balance between these two processes shifts toward increasing lung injury. This phase transition is also accompanied by an accelerated release of peptide-free elastin-specific desmosine crosslinks as the mean alveolar wall diameter begins to increase, suggesting their potential use as a biomarker for the molecular changes that precede the development of pulmonary emphysema. Early detection of the disease would allow more timely therapeutic intervention involving multiple agents that address the complexities of emergent phenomena at different scale levels.

## 1. Introduction

The pathogenesis of pulmonary emphysema encompasses multiple mechanisms involving numerous lung components. However, a critical component of the disease is the degradation of lung elastic fibers and their core elastin protein [1]. Inflammatory cells recruited to the lung by cigarette smoke and other toxic agents produce enzymes and oxidants that disrupt elastic fibers, thereby impeding the transmission of mechanical forces responsible for breathing. This process increases alveolar wall strain, leading to the rupture of airspaces and reduced gas exchange.

The recognition of excess elastase activity as a cause of airspace enlargement is based on alpha-1 antitrypsin (AAT) studies, which revealed that a deficiency of this protease inhibitor was associated with pulmonary emphysema [2]. Subsequently, other investigators demonstrated that AAT is an inhibitor of neutrophil elastase, an enzyme that induces emphysema when instilled in animal lungs [3]. The use of elastases to induce pulmonary emphysema remains an essential tool for investigating the molecular and morphological changes that contribute to this disease, although other models involving cigarette smoke and other agents have subsequently been developed [4].

While the relationship between inflammation and airspace distention plays an important role in the development of pulmonary emphysema, an additional mechanism may contribute more directly to the architectural changes associated with the disease. The uneven distribution of the mechanical forces may be necessary to convert proteolytic injury into airspace enlargement [5]. This concept is supported by in silico studies demonstrating that uneven mechanical strain initially produces localized distention of alveolar walls that evolve into widespread airspace enlargement resembling that seen in pulmonary emphysema [6]. The potential importance of this mechanism is supported by studies showing that overdistention of elastic fibers increases their susceptibility to enzymatic breakdown [7]. This finding emphasizes the limitations of focusing exclusively on elastases as the sole cause of alveolar wall injury.

The complex interaction between inflammatory processes and mechanical forces is consistent with an emergent phenomenon, where multiple scale-level interactions cause unpredictable systemic changes [8]. An example of this type of behavior is an epidemic, in which disease transmission depends on indeterminate effects of multiple factors [9,10]. As a result, the pattern of infection may remain uncertain until the disease involves a relatively large population.

The principle of emergence arose from the realization that physical phenomena cannot be explained entirely in terms of their individual parts [11]. Instead, the interaction of the system components can lead to behaviors not predicted by their individual properties. An example of this process is a chemical reaction in which combining molecules to form new products may involve indeterminate factors such as concentration, temperature, and time. This concept was applied to the lung when it was suggested that pulmonary disease cannot be fully understood by studying the pathogenetic mechanisms in isolation [12].

As an extension of this approach to understanding the pathogenesis of pulmonary emphysema, the current paper proposes that the emergence of the airspace enlargement involves the dynamic interplay between mechanical strain and elastin crosslinking. Based on our previous findings in animal models of the disease and human emphysematous lungs, we hypothesize that the tension between these two opposing components significantly contributes to the transition to an active disease state that is more resistant to treatment. This concept has practical implications for the early detection and treatment of pulmonary emphysema, which could significantly reduce the risk of respiratory failure.

## 2. Emergent Phenomena in the Pathogenesis of Pulmonary Emphysema

A fundamental aspect of emergent phenomena involves competing forces that shape the dynamics within a system. In pulmonary emphysema, the emergent phenomenon of airspace enlargement illustrates how complex relationships between mechanical stress and inflammatory processes culminate in significant alterations to lung architecture and function. As the uneven distribution of mechanical forces increases due to hyperinflation and loss of elastic recoil, they create added strain on alveolar walls. This mechanical stress can enhance the expression of various cytokines, which recruit inflammatory cells to the injury site [7]. These cells produce proteolytic enzymes and reactive oxygen species, which further degrade the extracellular matrix and worsen alveolar destruction. This mechanism creates a positive feedback loop in which mechanical damage enhances the inflammatory response, and the resulting inflammation leads to further mechanical compromise.

## 3. The Nonlinear Relationship Between Elastin Crosslinking and Airspace Enlargement

The dynamic interplay between elastin crosslinking and mechanical strain can lead to significant emergent phenomena at structural and functional levels. Elastin crosslink density can alter tissue biomechanics, affecting how cells respond to strain. When elastin becomes overly crosslinked, the mechanical properties of the alveolar wall change, leading to stiff, less compliant tissue that cannot effectively accommodate mechanical strain [13] (Figure 1). By identifying the mechanisms involved in this process, it may be possible to develop targeted interventions that maintain alveolar architecture. For example, therapies that change elastin crosslinking dynamics could enhance the alveolar wall’s resilience to mechanical strain, potentially slowing the emergence of an active disease state that is less amenable to treatment.

To further understand the interaction between elastin crosslinking and mechanical forces in pulmonary emphysema, our laboratory investigated the potential relationship between airspace size and the level of the elastin-specific crosslinks, desmosine and isodesmosine (DID) in emphysematous and normal human postmortem lungs (Figure 2). The lung content of peptide-free DID and the crosslink density in formalin-fixed, paraffin-embedded tissues were measured using a combination of liquid chromatography and tandem mass spectrometry [14]. The results showed a marked increase in free lung DID when the mean alveolar diameter exceeded 400 µm (Figure 3).

In response to the accelerated breakdown of elastin, there was a marked increase in DID density at 300 µm, which plateaued at 400 µm (Figure 4). The surface area of elastic fibers also peaked at a similar alveolar diameter, but to a lesser degree than DID density, consistent with a preferential increase in crosslinking rather than a nonspecific proliferation of the fibers. These findings support the hypothesis that airspace distention is an emergent phenomenon characterized by early proliferation of DID crosslinks that progress to a phase transition consisting of a rapid increase in elastin breakdown, accelerated rupture of alveolar walls, and development of an active disease state less amenable to therapeutic intervention.

The role of crosslinks in preserving the architectural properties of the lung was demonstrated by using β-aminopropionitrile (BAPN), a crosslink inhibitor, in a model of cadmium chloride-induced lung injury [15]. Animals receiving BAPN showed emphysematous changes, whereas those left untreated developed pulmonary fibrosis. This finding emphasized the role of DID in the pathogenesis of pulmonary emphysema.

In contrast to our findings, earlier investigations have shown that pulmonary emphysema is associated with unchanged or reduced DID content compared to normal lungs [16,17,18]. This disparity may be due to their use of lungs with more advanced disease [18]. The increase in DID that we observed may involve the formation of additional DID by condensing bifunctional crosslinks or incorporating newly synthesized elastin peptides into existing elastic fibers [19,20].

## 4. Modeling the Relationship Between Elastic Fibers and Mechanical Strain

The effects of mechanical forces on alveolar walls were modeled using a three-dimensional network of interconnecting units (K1 and K2), which correspond to intact or fragmented elastic fibers, respectively [21]. The movement of mechanical forces through the network is dependent on the ratio of K1 to K2. When the ratio is high, these forces are mainly transmitted through K1 units, preventing lung architecture disruption. Conversely, when K2 units predominate, the movement of mechanical forces largely involves the weaker pathways of the network, resulting in distention and rupture of alveolar walls.

Although this model greatly simplifies the actual role of differential mechanical forces on lung elastic fibers of variable thickness and orientation, it nevertheless provides a useful visual and conceptual representation of the overall effect of these forces on the elastic fiber network, particularly with regard to elastin crosslinks. The unraveling and fragmentation of the damaged elastic fibers would increase their susceptibility to enzymatic breakdown, causing the further release of free DID (Figure 5). This process may be exacerbated by the uneven distribution of mechanical forces, which could increase the strain on DID, resulting in the separation of their lysyl side chains from the surrounding amino acids.

The rapid increase in free DID at an MLI of 400 µm is consistent with this process and reflects a phase transition to an active disease state with important implications for treating pulmonary emphysema. Before the phase transition, therapeutic agents that prevent elastic fiber injury may shift the balance between injury and repair, thereby maintaining the alveolar wall structure. However, the efficacy of such treatment could decrease significantly as alveolar wall rupture accelerates.

## 5. The Effect of Elastic Fiber Fragmentation on the Transmission of Mechanical Force

Mechanical force is analogous to the flow of electricity, where current density represents electrical charge per unit area, and force density is the intensity of a force acting on a given area. This analogy helps conceptualize how a force, like a pushing or pulling action, can be distributed across a surface. As shown in an earlier publication, the formula for current density may be used to describe the transmission of mechanical forces through the K1 and K2 units as follows [23]:*j* = *ρ*_1_*v*_1_ + *ρ*_2_*v*_2_
where *ρ*_1_*v*_1_ and *ρ*_2_*v*_2_ represent the force density in the K1 and K2 units, respectively [23]. Similarly, the impact of structurally modified fibers on this parameter can be characterized in terms of the factors that affect electrical conductance using the following formula:$$G=\frac{A}{\rho l}$$
where *G* is conductance, and *A*, *l*, and *ρ* represent the conducting material’s area, length, and resistivity, respectively [23].

The formula for conductance incorporates the basic variables of fluid transmission (analogous to force transmission) through connecting bonds. While this formula does not consider the characteristics of the tissue surrounding elastic fibers, it does relate the physical properties of the fibers per se to their capacity for transmission of forces.

Although current flow and mechanical force are not identical, their properties sufficiently overlap to justify the use of the formula with regard to elastic fibers, where their distention and fragmentation lead to an increase in length and a reduction in cross-sectional area [23]. These structural changes impede the transmission of mechanical forces and increase the strain on alveolar walls.

## 6. A Feedback Loop Involving Structurally Altered Elastic Fibers

A hamster model of pulmonary emphysema involving sequential intratracheal instillations of elastase and LPS was used to study the relationship between lung inflammation and elastic fiber injury [22]. To enhance the relative effect of LPS, a single low dose of elastase was given before administering LPS. This regimen increased the impact of LPS on airspace enlargement, facilitating the investigation of potential synergistic interactions between the two agents.

The study’s objective was to determine if pretreatment with a low dose of elastase was sufficient to modify the structure of elastic fibers, making them more susceptible to subsequent LPS-induced lung injury. The results indicated that the combination of elastase and LPS resulted in a significant increase in both the total number of leukocytes and the percentage of neutrophils in bronchoalveolar lavage fluid (BALF) compared to the groups receiving elastase and saline, saline and LPS, or saline alone. The animals given both elastase and LPS also showed a marked increase in airspace enlargement compared to those given either agent alone. Furthermore, they were the only group with significantly higher levels of free BALF DID.

The effect of elastin peptides derived from human lung elastic fibers was also investigated in an LPS model of acute lung injury [22]. The concurrent instillation of elastin peptides and LPS produced significant increases in the percentage of BALF neutrophils and free BALF DID compared to treatment with either agent alone.

This study was followed by in vitro experiments utilizing BALF macrophages derived from untreated hamsters to measure the chemotactic activity of elastin peptides [22]. When administered separately, elastin peptides and LPS increased macrophage chemotaxis relative to controls. However, combining the two agents significantly enhanced chemotaxis compared to either agent alone. These findings support the concept of a self-perpetuating process of alveolar wall injury involving elastin peptide-induced recruitment of inflammatory cells, enhanced elastic fiber degradation, and further propagation of these proinflammatory peptides.

## 7. The Effect of Increased Crosslink Density on the Structural Integrity of Elastic Fibers

While no single formula explicitly quantifies the relationship between elastin crosslinking and mechanical strength, various models and theoretical frameworks have been proposed in the literature to explore this complex relationship. Generally, the mechanical properties of elastin-rich tissues can be described using material science principles, such as stress–strain relationships, elasticity, and viscoelasticity [24].

One way to approach this is through empirical models, which often relate the mechanical strength of the tissue to the degree of crosslinking. A simplified version of this relationship might be represented as follows:$$\sigma =kC^{n}$$
where

σ = mechanical strength of the tissue (e.g., tensile strength);

*C* = degree of crosslinking (this could be a measure of the density or number of crosslinks);

*k* = a constant that reflects the material properties of the specific tissue being examined;

*n* = an exponent that varies depending on the tissue and the specific biomechanical context (often found through experimental data).

According to this equation, as crosslinking increases, mechanical strength is also likely to increase, although the exact nature of that relationship can vary. For many biological materials, the exponent *n* is greater than one, indicating a nonlinear relationship where increases in crosslink density result in disproportionately higher increases in mechanical strength. Excessive crosslinking increases stiffness, reducing the distensibility of the fibers and making them more prone to fracture under mechanical stress [24]. This fracturing can compromise the structural integrity of alveolar walls, resulting in their distention and rupture.

The potential effect of mechanical forces on elastin synthesis is supported by in vitro studies of human myometrial cells, indicating that cyclical strain induces a significant increase in elastin synthesis [25]. Furthermore, repetitive mechanical force applied to an in vitro model of arterial tissue resulted in enhanced maturation of elastic fibers and an associated increase in DID content [26]. Interestingly, these findings suggest that mechanical forces may induce elastin proliferation in the same way exercise causes muscle cell hypertrophy. This repetition of mechanistic patterns at different levels of scale is characteristic of percolation processes [11].

## 8. The Role of Genetic Abnormalities

Although environmental factors, particularly cigarette smoking, play a dominant role in the development of this condition, genetic predispositions have surfaced as crucial factors that can exacerbate or modify the disease trajectory. In addition to the well-documented role of AAT deficiency in pulmonary emphysema, genetic variations in MMP genes, particularly MMP-12, have been implicated in emphysema [27]. Overexpression of these proteases can lead to excessive breakdown of extracellular matrix components in the lung, contributing to alveolar destruction and emphysema.

Other factors implicated in the pathophysiology of emphysema include genetic deficiencies that affect elastin crosslinking, which have drawn particular attention. Genetic mutations affecting these enzymes responsible for elastin and collagen synthesis can lead to various connective tissue disorders, including those associated with pulmonary conditions [28]. Furthermore, conditions such as Marfan syndrome, caused by mutations in the fibrillin-1 gene, can lead to altered elastin production and crosslinking [29]. Patients with this disorder are at a higher risk for developing pulmonary emphysema due to the fragility of lung elastin and resultant increased susceptibility to destruction.

Surfactant proteins, specifically SP-A and SP-D, reduce surface tension in the alveoli and have immunoregulatory functions. Genetic variations affecting surfactant protein metabolism may contribute to the pathogenesis of pulmonary emphysema by increasing susceptibility to lung injury and inflammation, thereby facilitating the development of airspace enlargement [30].

Other potential genetic variations that may affect the development of pulmonary emphysema include deficiencies that promote inflammatory pathways, resulting in sustained lung inflammation and structural changes that favor emphysema development. Genetic predispositions affecting antioxidant defenses can increase oxidative stress in lung tissues, damaging cells and contributing to airspace enlargement [31].

Although many of these genetic factors remain poorly understood, their effect on elastin crosslinking and airspace distention can significantly alter the natural progression of pulmonary emphysema and limit the efficacy of treatments designed to shift the balance between injury and repair. Moreover, potential synergistic interactions between genetic variations and environmental factors could exacerbate alveolar wall injury and further impair therapeutic intervention.

## 9. The Application of the Concept of Emergence to Therapeutic Intervention

As the pathophysiological processes of emphysema evolve nonlinearly, symptoms may not correlate directly with the extent of lung damage as assessed through imaging studies or pulmonary function tests. Recognizing that the interplay between mechanical and inflammatory factors can lead to unexpected functional declines emphasizes the need for comprehensive evaluation and individualized treatment strategies.

Understanding the emergent nature of airspace enlargement may also include identifying critical intervention points where therapeutic strategies could effectively disrupt the convergence of events at multiple levels of scale that lead to an active disease state. For instance, interventions that alleviate mechanical stress, such as lung volume reduction surgery, bronchoscopic lung volume reduction, or bronchodilators, may help stabilize lung mechanics and limit alveolar wall rupture [32,33]. Targeting multiple components of the emergent process, rather than a single inflammatory mechanism, may provide a more effective approach to therapeutic intervention.

While current efforts to develop a treatment for pulmonary emphysema have focused on elastase inhibitors, our laboratory has investigated using aerosolized hyaluronan (HA), a long-chain polysaccharide, to mitigate emphysematous changes. This approach is based on earlier studies showing that pretreatment with hyaluronidase exacerbates airspace enlargement induced by intratracheal elastase instillation [34]. In contrast, exposure to aerosolized HA in either elastase or cigarette smoke models of pulmonary emphysema has a protective effect, reducing alveolar wall distention and rupture [35,36]. This effect can be attributed to the ability of HA to bind to elastic fibers, creating a physical barrier against elastin-degrading agents (Figure 6). The therapeutic potential of supplementing the pulmonary extracellular matrix with exogenous HA is further supported by a study indicating a significant reduction in the level of this polysaccharide in the lungs of patients suffering from alpha-1 antitrypsin deficiency-related pulmonary emphysema [37].

The clinical efficacy of HA was studied in a 28-day study involving patients with pulmonary emphysema resulting from alpha-1 antiprotease deficiency [38]. Patients who inhaled HA twice daily exhibited a significant reduction in urine levels of free DID during the trial, supporting the concept that aerosolized HA effectively decreases elastin degradation and the associated release of proinflammatory elastin peptides [39]. Furthermore, HA may have therapeutic effects related to its hydrophilic characteristics, potentially enhancing energy storage within elastin and reducing the mechanical stress that leads to the fragmentation of elastic fibers [40,41,42]. This hypothesis is supported by a study demonstrating that HA and other proteoglycans maintain the normal distribution of mechanical forces in the pulmonary extracellular matrix [43].

The multifaceted effects of HA on various mechanisms of lung injury emphasize the necessity for developing comprehensive therapeutic strategies that address multiple inflammatory pathways. Aside from using alpha-1 antiproteinase in a limited group of COPD patients, interventions targeting a single inflammatory component have demonstrated limited effectiveness [44,45,46,47]. This finding may be attributed to the complex nature of biological systems, where systemic reorganization relies on numerous interactions across different scales. Treatments that focus on a single pathway may be circumvented by events at higher levels of scale, where the relationship between mechanical forces and alveolar wall distention supersedes individual molecular processes [48,49,50,51].

One of the critical convergence points in the pathogenesis of pulmonary emphysema may be the release of elastin peptides, which are both a consequence and a cause of elastin degradation due to their proinflammatory properties. Our work suggests that these peptides may act as proinflammatory agents in transmitting elastic fiber injury and airspace enlargement through the lung, like a disease vector operates in an epidemic [52].

Current research is focused on pharmacological agents that specifically inhibit the signaling activity of these peptides by their attachment to the elastin receptor complex (ERC) on inflammatory cells. Potential treatments include monoclonal antibodies that disrupt elastin peptide interactions with the ERC and gene therapy aimed at downregulating receptor expression [53].

The evaluation of novel drugs for treating pulmonary emphysema would be significantly improved by real-time assessment of their effectiveness. Clinical trials currently rely on pulmonary function studies, which require considerable time to reveal a therapeutic effect [54]. High-resolution computed tomography offers a more sensitive means of determining a successful outcome but also involves a prolonged interval to demonstrate positive results [55].

Multiple inflammatory mediators have been proposed as biomarkers for pulmonary emphysema [56,57]. However, free DID crosslinks may better indicate this disease because they reflect structural alterations in elastic fibers associated with alveolar wall injury. Although co-existing conditions such as osteoarthritis and atherosclerosis might reduce the specificity of blood and urine measurements, the DID levels in these fluids could still be a meaningful endpoint in clinical trials. As shown in our trial of aerosolized hyaluronan, differences in crosslink levels between closely matched experimental and control groups could provide a real-time measure of therapeutic efficacy [38]. Furthermore, the use of tandem mass spectrometry to accurately measure small amounts of free DID in sputum, BALF, breath condensate, and transbronchial or thoracoscopic biopsies would enhance the sensitivity and specificity of this biomarker for alveolar wall injury [58].

## 10. Conclusions

The concept of emergent phenomena provides a valuable framework for understanding the complex interactions between mechanical stress and inflammatory processes in pulmonary emphysema. Further investigation of feedback loops and nonlinear dynamics in the emergence of the disease may permit the design of more effective therapeutic strategies to slow the progression of alveolar wall injury. This process will involve detecting the initial patterns of lung injury before the disease undergoes a phase transition that is less amenable to treatment.

Although the idea of emergence in disease development is not new, investigating this phenomenon has been limited by the difficulty of relating specific biochemical events to morphological and physiological changes. In the current paper, these studies were limited to DID and alveolar diameter measurements. However, more refined techniques, such as computerized multiscale simulations of the molecular and mechanical mechanisms involved in the pathogenesis of pulmonary emphysema, may permit a more comprehensive understanding of the emergent properties of the disease. As these investigations begin to reveal critical changes in the molecular structure of alveolar walls, it may be possible to identify more sensitive and specific biomarkers that detect the disease at an early stage and permit more timely therapeutic intervention.

## Figures and Tables

**Figure 1 cells-14-00702-f001:**
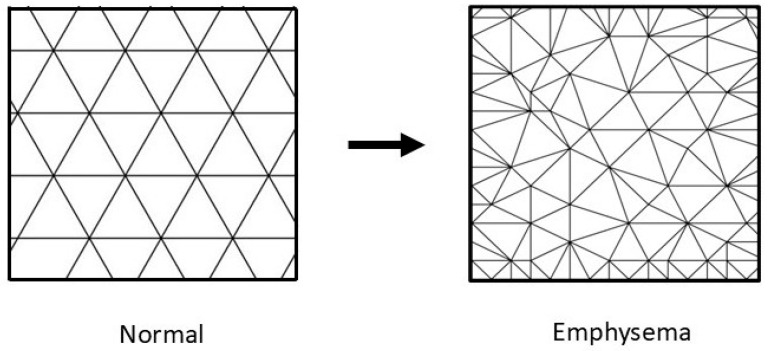
Illustration showing the increase in elastin crosslinking (intersecting lines) following repeated injury and repair of elastic fibers. The haphazard nature of this process results in uneven distribution of mechanical forces, which contributes to alveolar wall injury.

**Figure 2 cells-14-00702-f002:**
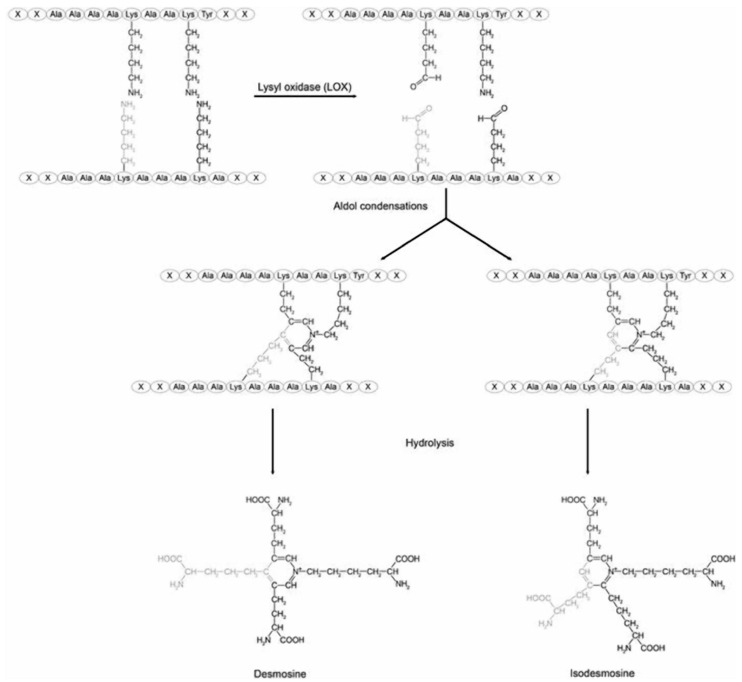
The desmosine and isodesmosine crosslinks of elastin are formed by the condensation of four lysine residues on adjacent peptides. The difference between them is the location of the lysyl side chains on the central pyridinium ring. Reprinted with permission of Creative Commons (https://creativecommons.org/licenses/by-sa/4.0/ (accessed on 10 February 2025)).

**Figure 3 cells-14-00702-f003:**
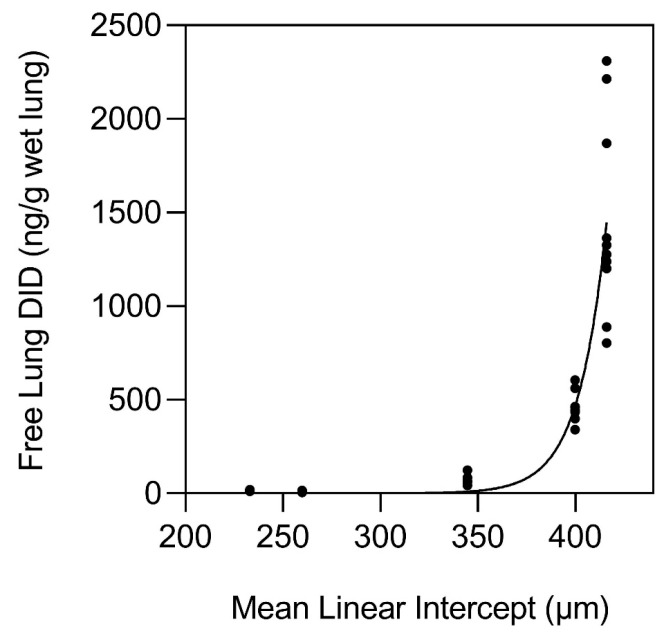
Graph showing the marked increase in lung content of free DID as the alveolar diameter exceeds 400 µm. This finding is consistent with the nonlinear features of emergent phenomena. Reprinted with permission from [14].

**Figure 4 cells-14-00702-f004:**
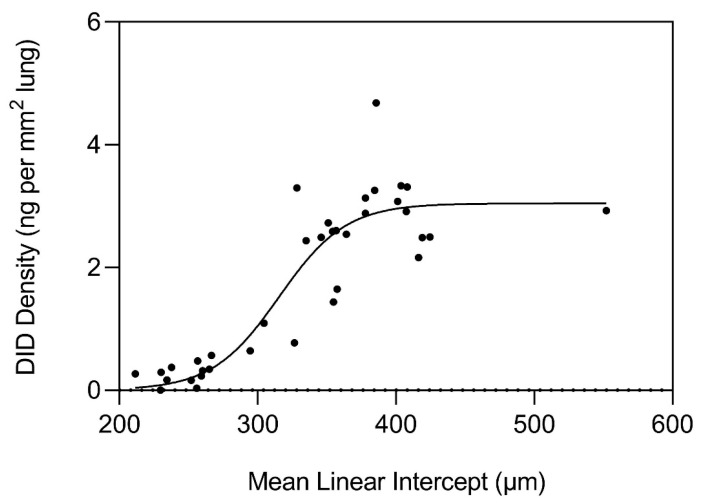
Graph showing the marked increase in DID density as alveolar diameter exceeds 300 µm. Beyond 400 µm, the density plateaus as the repair process undergoes a decompensatory phase due to progressive distention and rupture of alveolar walls. Reprinted with permission from [14].

**Figure 5 cells-14-00702-f005:**
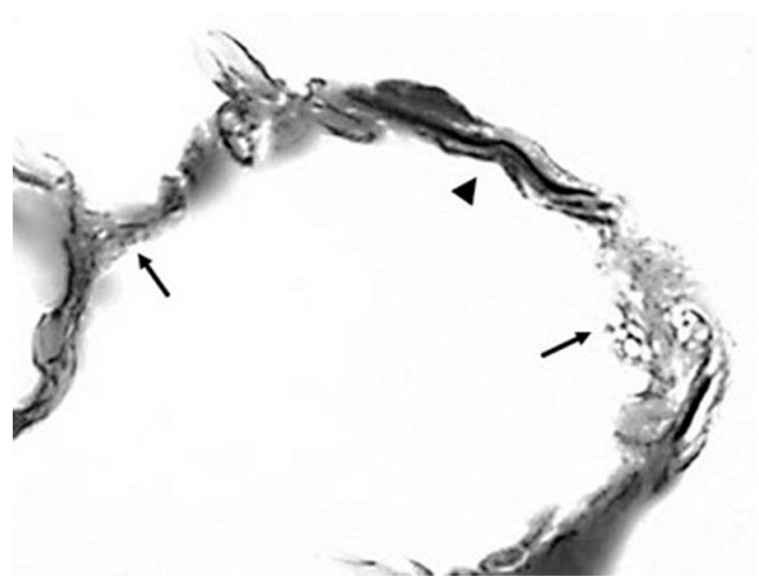
Photomicrograph showing the fragmentation (arrows) and unraveling of elastic fibers (arrowhead). Reprinted with permission from [22].

**Figure 6 cells-14-00702-f006:**
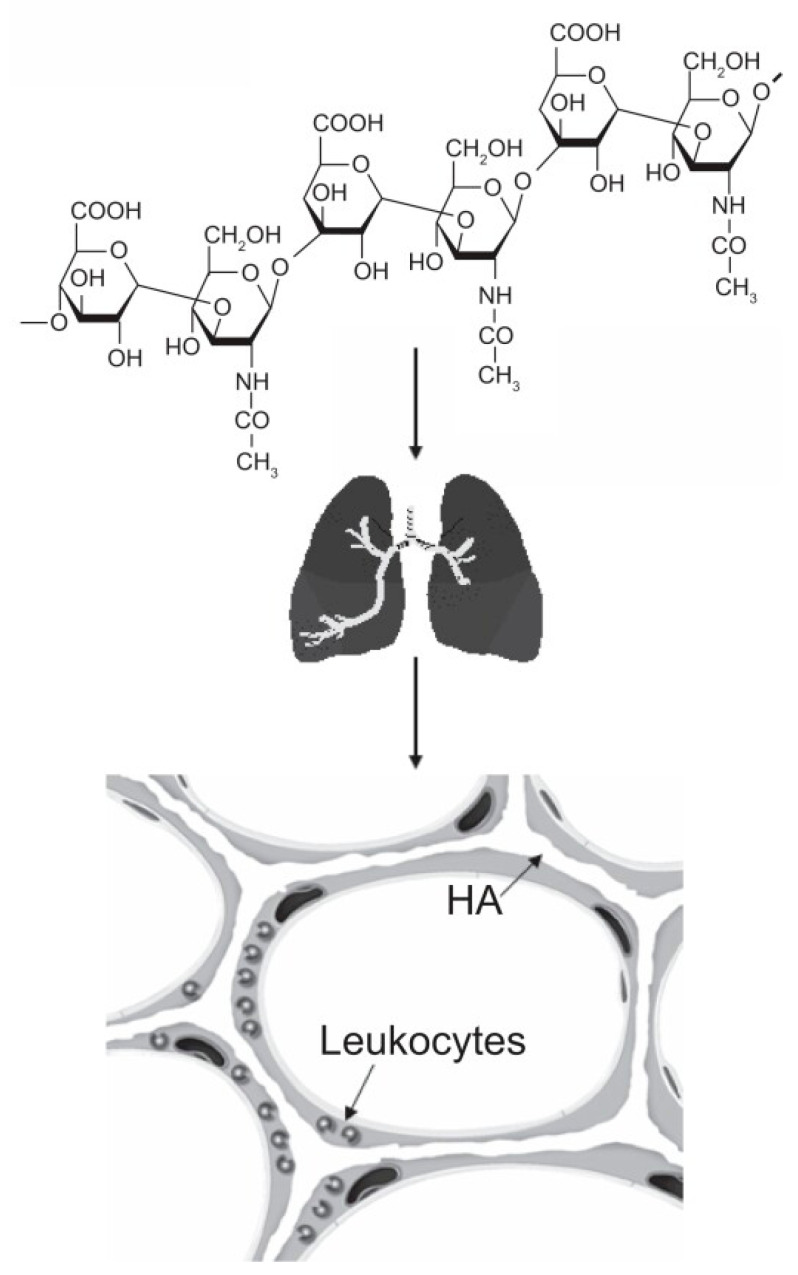
HA is a long-chain polysaccharide composed of repeating disaccharide units of glucuronic acid and N-acetylglucosamine. When inhaled as an aerosol, it binds to alveolar elastic fibers and protects them from elastases released by inflammatory cells. Reprinted with permission from [36].

## Data Availability

No new data were created or analyzed in this study.

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
