# Peer review of "The Competitive Interaction of Alveolar Wall Distention with Elastin Crosslinking: A Mechanistic Approach to Emergent Phenomena in Pulmonary Emphysema"

_cells, 2025, doi:10.3390/cells14100702_

Round 1

Reviewer 1 Report

Comments and Suggestions for Authors

The authors' point that in chronic respiratory diseases involving airspace remodeling, multi-scale mechanical forces interact in a complex manner and emergent phenomena arise. And I believe that illuminating this point would be beneficial to the field. However, there are major points the authors should address to make this manuscript more compelling. 

line #75-78, Figure 1 legend 
Although it is understandable that the authors want to express that crosslinking is uneven in emphysema in Figure 1, it should be corrected that the figure shows that the tissue density is higher in emphysema than in normal. Emphysema is basically a deliterious disease with a decrease in fibers, but the figure shows that the fiber density is higher to the extent that it is reminiscent of fibrosis or scar formation. At least in the emphysema figure it should not be drawn in a way that it seems to have more fiber than in the normal figure. 

line #150-164
I have observed a significant degree of similarity between this section and the content presented in reference #19. While the authors have cited reference [19] twice within this section, I respectfully suggest that all sentences within this section should be appropriately attributed to [19] to ensure accurate and comprehensive citation.
Furthermore, I have identified a potential discrepancy in the interpretation of the electrical conductance formula as it relates to the mechanical stress on the fiber networks. Specifically, both the current manuscript and reference [19] appear to interpret the area 'A' in the formula differently. My understanding is that 'A' typically denotes the cross-sectional area of a conducting material. However, the authors appear to interpret it as the surface area within the lung, which represents a distinct concept.
Additionally, I would like to inquire about the authors' rationale for primarily relying on the electrical conductance formula to describe mechanical stress on the alveolar wall. There exists a substantial body of literature dedicated to formulating mechanical strain in this context. I am curious as to why these relevant literatures were not consulted in favor of the electrical conductance formula, which seems less directly applicable to the mechanical stress under consideration.

Author Response

Response to Reviewer Comments

Textual revisions to the manuscript are highlighted in grey

Reviewer 1:

  1. As requested, the diagram of altered elastic fibers in pulmonary emphysema is now represented by lines with the same density as normal fibers (figure 1).
  2. The manuscript has been revised to include additional citations of the appropriate reference to the relevant formula (reference 22, formerly 19; p 8).
  3. We agree that the area (A) in the formula for conductance (G) is analogous to cross-sectional area of elastic fibers, not surface area, and have revised the text accordingly (p 8).
  4. The use of a current density formula instead of one related to mechanical strain is based on the random resistor model describing the weakening of the elastic fiber network. This model involves a loss of conductance due to random breakage of conducting bonds which is analogous to elastic fiber fragmentation.  While formulas for mechanical strain may be more appropriate to express changes in alveolar wall strain, we believe that the analogy to current density is more consistent with our model involving changes in force density within elastic fibers. We have included this explanation in the revised manuscript (pp 7-8).

Reviewer 2 Report

Comments and Suggestions for Authors

I congratulate the author for the perspective he presents regarding the aspects related to the pathophysiological mechanisms and the solutions that medicine offers today to patients with pulmonary emphysema. However, I believe that the perspective is not complete if it does not also address the role of the genetic factor both in the development of emphysema and in the limitations that previous research has. Certainly, completing the information with these data would bring the presented perspective into a specific topicality of the times.

Author Response

Response to Reviewer Comments

Textual revisions to the manuscript are highlighted in grey

Reviewer 2:

  1. In response to the reviewer comments, we have included a section on genetic factors that may have a poorly understood influence on the development of pulmonary emphysema and have added the appropriate references (pp 9-10).

Reviewer 3 Report

Comments and Suggestions for Authors

In this manuscript, the author argues that a critical component in explaining  the architectural changes associated with the development of pulmonary emphysema, is the uneven distribution of the mechanical forces which convert the initial proteolytic injury, due to the presence of heightened leukocyte inflammation, into airspace enlargement. The author provides in vitro data showing that as airspace size increases, the unique elastin crosslinks, desmosine and isodesmosine, increase concomitantly. However, the author notes that excessive crosslinking increases stiffness and reduces the distensibility of the fibers, making them more prone to fracture under mechanical stress.
While the manuscript introduces an interesting hypothesis on the development of human emphysema, however, there are several areas that require refinement:
1.    Clarification of the Manuscript's Objective: It is unclear if this a hypothesis driven manuscript or a narrative review. Therefore, this distinction should be reflected in both the title and content.
2.    Enhancement of the Introduction: The introduction requires substantial improvement. It should contextualize the hypothesis within the existing body of literature and discuss in more detail current models of pulmonary emphysema development.
3.    Addressing the Limitations of Current Models: Evidence  from prior studies should be included to demonstrate that the model of purely elastic fiber degradation does not fully account for the pathology of emphysema. This will help underline the novelty of the proposed hypothesis.
4.    Clear Differentiation of Existing Knowledge and Novel Contributions: There should be a clear distinction of what is already established in the literature from the novel contributions of the current work.

Author Response

Response to Reviewer Comments

Textual revisions to the manuscript are highlighted in grey

Reviewer 3:

  1. The revised manuscript includes a more complete discussion of the objectives of the manuscript and introduces the main hypothesis regarding the interplay of elastn crosslinking with alveolar wall distention. In addition, we have incorporated this concept into the manuscript title (pp 1-2).
  2. As requested, we have expanded the Introduction to include a historical perspective on the development of the protease-antiprotease model of pulmonary emphysema and how that led to the development of animal models of emphysema (p 1).
  3. The Introduction now includes a discussion of how uneven mechanical forces increase the susceptibility of elastic fibers to enzymatic breakdown, emphasizing the limitations of focusing exclusively on elastases as the sole cause of alveolar wall injury (pp 1-2).
  4. The revised manuscript includes a paragraph distinguishing earlier findings from the currently proposed interplay between elastin crosslinking and alveolar wall distention, which may have important consequences for the early detection and treatment of pulmonary emphysema (pp 1-2).

Reviewer 4 Report

Comments and Suggestions for Authors

This review article integrates the author's previous research findings to propose a theoretical framework of pulmonary emphysema pathogenesis. The manuscript presents the competition between mechanical strain and elastin crosslinking as a critical factor in disease progression, offering a valuable perspective. However, several aspects require refinement to enhance its scholarly contribution and clarity.

1. The current title accurately describes the content but fails to reflect that this is primarily a theoretical integration of previously published data rather than new experimental findings. The introduction should explicitly state that this manuscript aims to suggest a theoretical framework based on previously published findings to properly set reader expectations.

2. The author has missed a valuable opportunity by failing to cite two highly relevant self-published works: Diagnostics (Diagnostics, 2025 Feb 27;15(5):578.)(https://www.mdpi.com/2075-4418/15/5/578) and Life (Life. 2025 Feb 24;15(3):356.)(https://www.mdpi.com/2075-1729/15/3/356). These three papers form a complementary series examining emphysema from different angles: the current manuscript explores emergent phenomena and mechanical-biochemical competition; the Diagnostics paper focuses on DID as therapeutic biomarkers; and the Life article presents elastin peptides as disease vectors. By not cross-referencing these works, the author fragments what could be a cohesive theoretical framework. I recommend integrating these citations to demonstrate the evolution of the author's understanding across multiple dimensions of emphysema pathogenesis, significantly strengthening the manuscript's scholarly impact.

3. The frequent use of figures from previous publications raises copyright compliance concerns. Though citations are provided, it is essential to ensure proper permissions have been obtained and clearly documented. The author must provide written evidence of perpetual usage rights for figures originating from non-MDPI journals, as one-time permissions are insufficient for republication across multiple papers. The author should submit comprehensive permissions documentation as supplementary material to ensure academic transparency and copyright compliance.

4. The electrical network analogy (K1 and K2 units) significantly oversimplifies pulmonary biomechanics. These sections should include text cautioning readers about the limitations of the theoretical model. Critical limitations include: (1) The orientation and alignment of elastic fibers dramatically influence force distribution, yet the model treats fibers as isotropic elements. Even with identical fiber connectivity, differences in fiber orientation would result in fundamentally different mechanical responses. (2) Stress distribution varies considerably between diaphragmatic (abdominal) breathing and thoracic breathing, creating distinct mechanical environments that cannot be captured by the simplified electrical analogy.

5. The application of electrical formulas (G = A/ρl) to biological tissues neglects viscoelasticity, strain-dependent properties, and non-linear stress-strain relationships characteristic of pulmonary tissue.

6. In addition to using electrical formulas for simulation, high-performance computers with GPUs can now perform multi-scale simulations based on Newtonian mechanical principles rather than electrical circuit theory. The discussion section should propose molecular dynamics (MD) simulations as a method to validate the theoretical framework. Modern MD software like NAMD, combined with enhanced sampling techniques such as metadynamics, can now simulate systems at scales relevant to the author's hypotheses regarding elastin crosslinking and mechanical strain.

7. The author assumes free DID increases result from enzymatic degradation of crosslinked elastin. Consider an alternative: inflammation and alveolar distention could physically release uncrosslinked DID molecules trapped within the elastin network. As alveolar walls stretch, intermolecular spaces widen, potentially liberating non-crosslinked DID without enzymatic activity. This perspective could explain the sudden increase in free DID at specific diameter thresholds. This possibility should be mentioned in the article if there is no evidence to exclude it.

8. Although the author's previous article (Diagnostics, 2025 Feb 27;15(5):578)(https://www.mdpi.com/2075-4418/15/5/578) discusses detecting DID as therapeutic biomarkers using blood or bronchoalveolar lavage fluid, the current manuscript lacks practical recommendations for tissue sampling methods to verify alveolar elastic fiber breakage or arrangement in the proposed theoretical model. Including suggestions for bronchoscopic or thoracoscopic biopsies would enhance clinical applicability. Additionally, discussing the feasibility of detecting elastin crosslinks in blood samples would provide valuable insights for potential non-invasive diagnostic approaches.

Author Response

Response to Reviewer Comments

Textual revisions to the manuscript are highlighted in grey

Reviewer 4:

  1. In response to the reviewers suggestions about revising the Introduction, we have framed the current paper in terms of our hypothesis regarding elastin crosslinking and alveolar wall distention, which is based on previous findings from our laboratory (p 2). The title of the paper has also been revised to reflect the hypothetical nature of review.
  2. We have added the two author references cited by the reviewer and indicated their relevance to detecting and treating alveolar wall injury in pulmonary emphysema (p 12; refs 50, 56).
  3. The reprint permission forms for the relevant figures have been uploaded as a supplemental file.
  4. The authors appreciate the simplicity of the percolation model of elastic fiber injury. It was designed to provide a minimal visual and conceptual framework for understanding the relationship between mechanical forces and elastic fiber injury.  This caveat is included in the revised manuscript (p 7).
  5. As stated in the response to Reviewer 1, the rationale for using the formula for current density instead of one related to mechanical strain is based on the random resistor model of structural changes in the elastic fiber network. This model involves a loss of conductance due to random breakage of conducting bonds and is frequently applied to studies of emergent phenomena, particularly those involving a nonlinear phase transition similar to that associated with the accelerated release of free DID in pulmonary emphysema.  While formulas incorporating the variables associated with mechanical stress may be more appropriate to express overall changes in alveolar wall strain, we believe that the application of the current density formula is more reflective of analogous changes in the force density within elastic fibers.  We have included this explanation in the revised manuscript (pp 7-8).
  6. The formula for conductance incorporates the basic variables of fluid transmission (analogous to force transmission) through connecting bonds; namely the physical characteristics of the conducting medium. While this formula does not consider the characteristics of the tissue surrounding elastic fibers, it does relate the physical properties of the fibers per se to their capacity to conduct forces. With regard to mechanics, the concept of force density is analogous to current density. In the same way that current density represents the flow of electrical charge per unit area, force density represents the intensity of the force on a given area. This analogy helps conceptualize how a force, like a pushing or pulling action, can be distributed over a surface area. While we acknowledge that the two forces are not identical, their properties sufficiently overlap to justify the use of the formula for conductance. This point is discussed in the revised manuscript (p 8).
  7. We agree with the reviewer that mechanical forces may dislodge elastin crosslinks in combination with elastases. This possibility is discussed in the revised manuscript (p 7).
  8. A discussion of the role of the DID biomarker has been added to the manuscript and includes the potential use of various matrices for measurement, such as bronchoalveolar lavage fluid, sputum, breath condensate, and transbronchial or thoracoscopic biopsies. In addition, the limitations of using systemic body fluids such as blood are discussed in terms of their contamination with DID from other sites such as the cardiovascular system and joints (p 12).

Round 2

Reviewer 1 Report

Comments and Suggestions for Authors

I appreciate the authors' efforts to address the points I raised in the previous round.

Author Response

Response to Reviewer Comments

Note: Textual revisions to the manuscript are highlighted in grey

Reviewer 1:

No additional comments

Reviewer 3 Report

Comments and Suggestions for Authors

In this manuscript, the author argues that a critical component in explaining the architectural changes associated with the development of pulmonary emphysema is the uneven distribution of the mechanical forces, which convert the initial proteolytic injury, due to the presence of heightened leukocyte inflammation, into airspace enlargement.
 The author provides in vitro data showing that as airspace size increases, the unique elastin crosslinks, desmosine and isodesmosine, increase concomitantly. However, the author notes that excessive crosslinking increases stiffness and reduces the distensibility of the fibers, making them more prone to fracture under mechanical stress.

In the revised manuscript, the author has provided additional data that further support this hypothesis. 
However, two important issues remain:
1. Potential mechanisms that lead to the increased elastin crosslinking (proliferation of DID crosslinks) in human emphysema have not been clearly defined. Potential mechanisms include increased work of breathing in the early stages of the disease that could lead to repeated injury and repair, and the excess of elastase and other inflammatory molecules that may drive the proliferation of DID. Please explain in depth and provide relevant studies.  

2. It is not clear if the proposed model is solely based on animal models and in vitro data, which would render it a hypothetical construct until validated by human studies. You need to provide a clear statement addressing this distinction.

Author Response

Response to Reviewer Comments

Note: Textual revisions to the manuscript are highlighted in gray.

Reviewer 3:

  1. In response to the reviewer’s question about the sources of increased crosslinking of elastin, we describe the roles of elastic fiber repair, mechanical stress, and conversion of bifunctional crosslinks to desmosine as potential mechanisms responsible for the increased DID density that could contribute to elastic fiber rigidity (pp 6, 9, refs 11, 19, 20, 25, 26).
  2. We have added statements indicating the use of animal models and human postmortem lungs to formulate our hypothesis (pp 2, 3).

Reviewer 4 Report

Comments and Suggestions for Authors

The authors have addressed nearly all of the comments satisfactorily, clarifying the theoretical nature of the manuscript in both the title and introduction, incorporating the suggested citations, and adding appropriate caveats to their modeling framework. They have also discussed alternative mechanisms for DID release and expanded on biomarker sampling strategies. The overall revisions are acceptable, and the manuscript is now suitable for publication.

Author Response

Response to Reviewer Comments

Note: Textual revisions to the manuscript are highlighted in grey

Reviewer 4:

No additional comments.

Round 3

Reviewer 3 Report

Comments and Suggestions for Authors

All issues have been satisfactorily addressed.